# Transcriptome Analysis to Explore the Cause of the Formation of Different Inflorescences in Tomato

**DOI:** 10.3390/ijms23158216

**Published:** 2022-07-26

**Authors:** Yahui Yang, Tingting Zhao, Xiangyang Xu, Jingbin Jiang, Jingfu Li

**Affiliations:** Laboratory of Genetic Breeding in Tomato, Key Laboratory of Biology and Genetic Improvement of Horticultural Crops (Northeast Region), Ministry of Agriculture and Rural Affairs, College of Horticulture and Landscape Architecture, Northeast Agricultural University, Harbin 150030, China; yyh201296@163.com (Y.Y.); 15504506671@sohu.com (T.Z.); xuxy@neau.edu.cn (X.X.)

**Keywords:** tomato inflorescence, differentially expressed genes, pathway

## Abstract

The number of inflorescence branches is an important agronomic character of tomato. The meristem differentiation and development pattern of tomato inflorescence is complex and its regulation mechanism is very different from those of other model plants. Therefore, in order to explore the cause of tomato inflorescence branching, transcriptome analysis was conducted on two kinds of tomato inflorescences (single racemes and compound inflorescences). According to the transcriptome data analysis, there were many DEGs of tomato inflorescences at early, middle, and late stages. Then, GO and KEGG enrichments of DEGs were performed. DEGs are mainly enriched in metabolic pathways, biohormone signaling, and cell cycle pathways. According to previous studies, DEGs were mainly enriched in metabolic pathways, and FALSIFLORA (FA) and ANANTHA (AN) genes were the most notable of 41 DEGs related to inflorescence branching. This study not only provides a theoretical basis for understanding inflorescence branching, but also provides a new idea for the follow-up study of inflorescence.

## 1. Introduction

Higher plants exhibit various inflorescence architectures progressing in complexity from a solitary flower to complex structures that contain multiple branches and flowers [1]. The architecture of inflorescences is one of the determinant traits for many crops, such as rice, maize, and tomato [2]. Favorable inflorescence branching is always a major breeding target for achieving desirable production by balancing the sink–source relationship [3]. Unlike the inflorescences of *Arabidopsis*
*thaliana* and rice, the inflorescences of tomato are cymes [4]. The SIM (sympodial inflorescence meristem) generates a new SIM before terminating in the FM (floral meristem), and the tomato inflorescence and branches are determined by the repetition of this pattern [5]. A series of regulatory genes that have received much attention have made major contributions to inflorescence architecture in tomato by changing the inflorescence branching pattern. In *Arabidopsis thaliana* and tomato [6,7,8,9,10], WUSCHEL-CLAVATA (WUS-CLV) feedback regulatory loops maintain a balance between SAM activities and control the size of the meristem [11,12]. Mutations in the CLV pathway genes, SlCLV3, FASCIATED AND BRANCHED (FAB), and FASCIATED INFLORESCENCE (FIN), cause meristems to enlarge, leading to an increase in inflorescence branching in tomato [13,14]. In addition, flowering time genes also play a certain role in regulating the inflorescence structure during the flowering period transition. The single-flower truss (SFT) is a flowering mutant that not only flowers late, but also disrupts the normal sympodial growth of tomato [15]. A small number of inflorescences produced by tomato SFT mutant plants under specific growth conditions can promote the transition of inflorescences to vegetative functions [16,17]. JOINTLESS (j) mutants produce indeterminate inflorescences and resume vegetative growth after producing 2–3 flowers [18]. In addition, J and SFT synergistically regulate the inflorescence structure to prevent early changes in inflorescence meristem (IM) characteristics after inflorescence morphogenesis [19]. In addition, tomato FALSIFLORA (FA) controls the flowering time and floral meristem characteristics [20]. FA mutations result in the transformation of flowers into secondary buds and the production of highly branched inflorescences [21]. In addition to these genes promoting flowering, TMF (TERMINATING FLOWER) also encodes an ALOG family protein that affects tomato inflorescence tissues [22]. The synergistic effect of TMF and slbop can prevent early flowering and increase inflorescence complexity. All inflorescences of slbop1/2/3 triple mutants only develop one or two flowers [23]. In addition, floral meristem characteristic genes also have an important influence on tomato inflorescence structure [1]. The absence of ANANTHA (AN) and COMPOUND INFLORESCENCE (S) leads to additional branching [24]. A heterozygous mutation combination of JOINTLESS2 (J2) and Enher-of -JOINTLESS2 (EJ2) can cause a certain number of inflorescence branches. A heterozygous mutation combination of JOINTLESS2 (J2) and Enher-of -JOINTLESS2 (EJ2) can cause a certain number of inflorescence branches [25,26,27]. However, due to the limited number of known inflorescence branching sites or genes and the unknown underlying mechanisms, the reconfiguration of inflorescence branching remains a challenge in tomato breeding for high-yield varieties. In higher tomato plants, inflorescence structure, flower number, fruit size and quality, and superior plant type are directly related to yield. The analysis of genes regulating related traits is of great significance for understanding the genetic basis of phenotypic variation and promoting the breeding of superior varieties. In this study, we compared the transcriptome between raceme and compound inflorescence (Racemes: rachis longer, unbranched, with many florets ascending from bottom to top; compound inflorescence: inflorescence rachis with branches, and each branch corresponds to a raceme), and the causes of different inflorescence branches of tomato inflorescence were analyzed through a large amount of data and the similarities and differences of genes controlling inflorescence branches were observed. According to previous studies, 41 genes related to the inflorescence branch were screened from DEGs. The presence of FALSIFLORA (FA) and ANANTHA (AN) genes among the 41 genes is the main reason for the different inflorescence branches in tomato. At the same time, plant hormones also affect the inflorescence branch. Gibberellin mainly affects tomato branches. We have been able to identify the main reasons for the different branches in tomatoes, both in terms of genes and hormones. It shows that transcriptome experiments can roughly control the formation process of compound inflorescence and lay a theoretical foundation for the subsequent study of compound inflorescence.

## 2. Results

### 2.1. Paraffin Sectioning of Tomato Inflorescences

We examined two different tomato inflorescence branches and paraffin sections. As shown in the figure below, two periods (early and middle) of paraffin sectioning were performed on compound inflorescence (CI) and single raceme (SR) material. As seen from Figure 1A, the flower branch sections tended to swell and increase, and the middle stage sections tended to sag inward. We can see from Figure 1B that in the middle stage of the flowers, both sides continued to enlarge, and the middle sag became obvious. We can see from Figure 1C that the inflorescence tomato slice was smooth and not obviously concave. Figure 1C,D show no significant differences.

### 2.2. RNA Sequencing Data and Functional Analysis

In this study, 18 samples were tested, and the average output of each sample was 6.72. The reference genome was 92.90% for the NCBI SL3.0 (USA, NIH, 1988) assembly sample and 80.75% for the comparison gene set. A total of 23,593 genes were detected. Illumina quality scores of 20 (Q20) and 30 (Q30) represented the percentage of sequencing data with error rates less than 1 and 0.1%, respectively. In this study, the clean reads ratio (%) was more than 94%, over 96% of reads met the Q20 threshold, while 91% of clean reads met the Q30 threshold (Table 1). We then compared clean reads with reference genome sequences using HISAT (USA, Johns Hopkins University), and more than 94% of clean reads were uniquely targeted reads.

### 2.3. Expression and Analysis of Differentially Expressed Genes

In CI and SR tomato plants, DEGs were selected with a Q value ≤ 0.05 and ∣log_2_FC∣ ≥ 1 (Figure 2). According to the figure, E_SR-vs.-L_SR up-regulated DEGs and down-regulated DEGs were the most obvious, among which 4554 genes were up-regulated and 3525 genes were down-regulated. Second, M_SR-vs.-L_SR up-regulated and down-regulated DEGs were more obvious, including 1947 up-regulated genes and 3094 down-regulated genes. However, in E_CI-vs.-E_SR, 732 DEGs were up-regulated and 352 DEGs were down-regulated. Moreover, in M_CI-vs.-M_SR, 887 DEGs were up-regulated and 1720 DEGs were down-regulated. Finally, in L_CI-vs.-L_SR, 1599 DEGs were up-regulated and 2436 DEGs were down-regulated. To more intuitively detect the changes in DEGs in different periods, we made volcano maps of DEGs in tomato inflorescences during the early, middle, and late periods. As shown in Figure 3, red dots represent up-regulated genes, green dots represent down-regulated genes, and gray dots represent non-DEGs.

### 2.4. GO Functional Classification Analysis of Differentially Expressed Genes

Through GO analysis, the biological functions of genes can be understood, as shown in Figure 4. The size of bubbles represents the number of genes annotated in the pathway. In general, Q ≤ 0.05 is thought to indicate significant enrichment. In the E_SR-vs.-E_CI group, genes were enriched in the ubiquitin ligase complex, chloroplast thylakoid lumen, cell wall, and chloroplast pathway (Figure 4A and Appendix A). In the M_SR-vs.-M_CI comparison, DEGs were significantly enriched in photosystem I reaction centers, membrane components, chloroplast thylakoid membranes, membranes, cell walls, and chloroplast stroma (Figure 4B and Appendix A). In the L_SR-vs.-L_CI comparison, DEGs were significantly enriched in the membrane components, extracellular region, cell wall, membrane, chloroplast outer membrane, and plasmodesmata pathway (Figure 4C and Appendix A). These results indicate that genes that regulate the number of inflorescence branches are mainly enriched in metabolic, cellular, and biological regulation pathways.

### 2.5. Classification of DEGs in KEGG Pathways

To better analyze the factors affecting inflorescence branching, we classified DEGs into five categories: cellular processes, environmental information processing, genetic information processing, and metabolic and organic systems. In the comparison of E_SR-vs.-E_CI, 14 DEGs were enriched in cell processes, 15 DEGs were enriched in environmental information processing, 28 DEGs were enriched in genetic information processing, 168 DEGs were enriched in metabolism, and 7 DEGs were enriched in organic systems (Figure 5A and Appendix A). In the M_SR-vs.-M_CI comparison group, 17 DEGs were enriched in cell processes, 10 DEGs were enriched in environmental information processing, 29 DEGs were enriched in genetic information processing, 75 DEGs were enriched in metabolism, and 11 DEGs were enriched in organic systems (Figure 5B and Appendix A). In the L_SR-vs.-L_CI comparison group, 55 DEGs were enriched in cell processes, 68 DEGs were enriched in environmental information processing, and 110 DEGs were enriched in genetic information processing. There was a metabolic enrichment of 473 DEGs and organic enrichment of 39 DEGs (Figure 5C and Appendix A). From the above description and Figure 5, we can see that most DEGs were enriched in metabolic pathways.

### 2.6. Screening of Genes Affecting Tomato Inflorescence Traits

By collating gene expression log2-fold change, KEGG, and GO data, we screened 41 candidate genes related to tomato inflorescence branches from the E_SR-vs.-E_CI and L_SR-vs.-L_CI groups (Table 2). Eight DEGs were associated with plant hormone signaling pathways. Among them, 3 DEGs, gibberellin-related gene GA2ox1, and auxin-related genes ARF3 and ARF4, were related to the flavonoid biosynthesis pathway, and 2 DEGs were related to the zeatin biosynthesis pathway. One DEG was associated with carbon metabolic pathways. Two DEGs were associated with cell cycle pathways. Six DEGs were associated with chloroplast synthesis. Five were associated with F-box synthesis. Two DEGs were associated with circadian pathways. Although the remaining 12 DEGs were not enriched in relevant pathways, they also played an important role in tomato inflorescence branching. For example, due to the loss of flower-forming ability, AN and FA led to the formation of the lateral synaxy meristem, resulting in the formation of cauliflower (an) and vegetative inflorescences (fa). In conclusion, some genes of the plant hormone pathway, circadian rhythm pathway, and cell cycle pathway, as well as genes without significant pathway expression changes, may be important reasons for the different branches of tomato inflorescence.

### 2.7. Gene Co-Expression Network Analysis

Weighted gene co-expression network analysis (WGCNA) is a tool for analyzing complex data. In particular, the co-expression network predicts the regulatory relationship between genes by using the expression correlation between genes. A dynamic shearing algorithm was used for gene clustering and module division, and finally, 13 gene co-expression modules were obtained, with the maximum number of genes in the modules being 9203 (honeydew) (Figure 6B). Each branch of the cluster tree represents a module. Each leaf represents a gene. Each color represents a module. Looking at the correlation coefficient of the modules, it was found that the genes in the dark red and yellow modules had higher specificity.

Therefore, we performed KEGG pathway analysis on the dark red and yellow modules. Deep red module genes were found to be enriched in photosynthesis, carbon fixation, carbon metabolism, chloroplast metabolism, carotenoid biosynthesis, and plant hormone signal transduction pathways during photosynthesis (Figure 6C). The genes in the yellow module were enriched in various pathways, including plant hormone signaling, the circadian rhythm, and the MAPK signaling pathway (Figure 6D).

### 2.8. MapMan Software Analyzes DEGs

MapMan (3.6.0RC1,Elijah Myers, USA) is a plant-specific software that provides a complete functional classification of genes and a comprehensive pathway map. The cell response, cell function, biological regulation, and metabolism of DEG in late inflorescence and late inflorescence of two tomato varieties were analyzed by using MapMan software. The red squares represent down-regulated differential genes and the blue squares represent up-regulated differential genes. The darker the color, the more significant the difference (Figure 7). In terms of the regulation of the cell response (Figure 7A), most DEGs involved in cell division and the cell cycle were down-regulated, and those involved in development were significantly up-regulated. In terms of the regulation of cell function (Figure 7B), there were more down-regulated genes than up-regulated genes in the cell division and cell cycle pathways. There were more up-regulated DEGs in the REDOX pathway, while there were more down-regulated genes in the DNA repair and DNA synthesis pathways. The results of the biological regulatory pathway (Figure 7C) showed that DEGs mainly affected indole acetic acid (IAA), abscisic acid (ABA), brassinosteroids (BR), ethylene, cytokinin (CTK), jasmonic acid (JA), salicylic acid (SA), and gibberellin (GA). Genes related to each hormone were also up-regulated and down-regulated. The content of DEGs in the IAA pathway was highest, and that in the salicylic acid pathway was lowest. From metabolic pathway analysis (Figure 7D), DEGs were enriched in the cell wall, lipids, secondary metabolism, amino acids, TCA, starch sucrose, and light sucrose. The expression of DEGs was up-regulated and down-regulated, and there were more down-regulated genes than up-regulated genes. Most of the genes were enriched in secondary metabolism, and there were more down-regulated genes than up-regulated genes.

### 2.9. Validation of RNA-Seq Data via qRT–PCR

To verify the accuracy of the DEG expression patterns indicated by the RNA-seq data, we analyzed 10 DEGs by qRT–PCR with three biological replicates (Figure 8). Relative quantitative data were calculated according to the CT method: normalization (CT = CT (sample) CT (GAPDH)); CT = CT (sample A) CT (sample B); relative quantification = 2 CT. The qRT–PCR data were analyzed with SPSS v.21.0 software for variance analysis, and the Waller–Duncan (W) method was used for comparison at the *p* < 0.05 level. It could be seen from the figure that under the background where *p* < 0.05 represents a significant difference, the data point marked ‘a’ was significantly higher than ‘b’ and ‘c’; ‘b’ was significantly higher than ‘c’. There was no significant difference between the same letter. These results indicated that the RNA-seq data were of high quality and could be used for subsequent analysis.

## 3. Discussion

In this study, we sequenced the early, middle, and late-stage inflorescences of two different tomato inflorescence branches. The transcriptome data were verified by qRT–PCR, and the results showed that the data obtained by RNA-seq were reliable. Transcriptome data showed that DEGs were abundant in the nine comparison groups. We set-up three comparison groups between the two materials: the early stage of the two materials, the middle stage of the two materials, and the late stage of the two materials. Therefore, we found that DEGs were the most abundant in the comparison of the early and late inflorescences, indicating that the genes regulating inflorescence were greatly expressed from the early to late inflorescence stages. We conducted GO and KEGG enrichment analyses for early and late DEGs and found that DEGs were significantly enriched in some effective pathways. To facilitate the further understanding of the gene functions, we also carried out co-expression analysis. Then, we used MapMan software to classify the biological functions of the DEGs, which made our study on the branching of inflorescences more detailed.

Our GO enrichment results revealed that the different branches of tomato inflorescences may be closely related to some pathways. In Figure 5, DEGs in different inflorescence branches were significantly enriched in the early stage, mainly in the ubiquitin ligase complex, chloroplast thylakoid lumen, membrane, cell wall, and chloroplast pathways. DEGs of different inflorescence branches were mainly involved in the photosystem I reaction center, membrane components, chloroplast thylakoid membrane, membrane, cell wall, and chloroplast stroma at the metaphase. DEGs in different inflorescence branches were significantly enriched in the membrane components, extracellular region, cell wall, membrane, chloroplast outer membrane, and plasmodesmata pathway at the late stage. We also sorted out DEGs from different inflorescence branches through the three main categories of molecular functions, cellular components, and biological processes. DEGs were found to be significantly enriched in metabolic processes, cellular processes, and biological regulation at early, middle, and late stages. According to the GO enrichment results, the different branches of inflorescence may be caused by the presence of genes in the chloroplast thylakoid membrane, photosystem reaction, membrane components, and membrane. At the same time, the branching is closely related to metabolic processes, cellular processes, and biological regulation.

We used MapMan to analyze DEGs from the four aspects of cell response, cell function, biological regulation, and metabolic regulation (Figure 8). From the aspects of cell response and cell function, we observed that the DEGs were significantly different in both the cell cycle pathway and the cell division pathway. Previous studies have shown that the cell cycle pathway can affect the growth of tomato fruits [28]. In this study, it was found that some genes located in the cell cycle pathway may also have a certain influence on the branching traits of tomato inflorescences, marking a new discovery with certain significance. Some studies have also shown that the cell division pathway has a certain influence on the growth of tomato throughout its life [29]. DEGs were found to be involved in auxin, cytokinin, and gibberellin pathways. Previous studies have found that auxin is a key regulatory factor in flower development [30]. When plants are treated with auxin polar transport inhibitors, the leaf sequence pattern changes, and lateral organs and floral meristems cannot be formed, thus forming a kind of acicular inflorescence [31,32]. As an important flowering hormone, cytokinins can regulate floral primordium initiation and floral organ development and participate in the regulation of stamen and pistil development of flowering plants, delaying the senescence of flower organs [33,34,35]. Gibberellin plays a role in regulating organ growth and promoting reproductive development in the whole growth process of crops [36]. Jasmonic acid signaling in crops is usually regulated by both biotic and abiotic stresses, but studies have shown that excess JAZ2 can promote tomato leaf growth and early flowering [37]. Metabolic pathway analysis of DEGs involved in the cell wall, lipids, secondary metabolism, amino acids, TCA, starch, sucrose, light reactions, photosynthesis, and nucleotides were enriched. Previous studies have shown that DEGs affect the growth of tomato stems through metabolic pathways [38]. According to the above findings, cell response, cell function, biological regulation, and metabolic regulation, both directly and indirectly affected tomato growth and development and inflorescence branching.

To better analyze the causes of different inflorescence branches in tomato, we selected effective genes from the transcriptome (Table 2). Previous studies have found that fruit germination after fertilization in angiosperms is strictly regulated by plant hormones and revealed that direct crosstalk between IAA and GA signaling components is crucial for tomato fruit formation [39,40]. ARF3 and ARF4 are members of the auxin response factor, ARF, family. The results showed that SlARF3 plays an important role in the formation of epidermal cells and trichomes and revealed the novel and specific functions of ARFs in tomato development [41]. Krp2 regulates or affects tomato fruit development through the cell cycle pathway [28]. CHS2 is involved in the circadian pathway and flavonoid biosynthesis pathway (FBP), which may be an ideal system for simulating genotypic and phenotypic interactions related to secondary metabolism [42]. Previous studies have reported that the heterozygosity of the SINGLE FLOWER TRUSS (SFT) allele for functional loss in tomato, the genetic origin of the flowering hormone floblast, can increase yield by 60%. The yield advantage of SFT heterozygosity is robust and occurs in different genetic backgrounds and environments. Studies have shown that the polymorphic integration of several traits drives heterosis in a multiplicative manner, and these effects are due to the inhibition of growth termination mediated by SELF PRUNING (SP). The results of this study provide the first example of the yield of a single hyperdominant gene and suggest that a single heterozygous mutation may enhance the productivity of other agricultural organisms [43]. SINGLE FLOWER TRUSS is a regulator of flowering time and branch structure encoding a tomato lineal homolog of FT, a major flowering integron gene in *Arabidopsis thaliana*. Grafting signals generated by SFT complement the morphogenetic defects of *sft* plants and replace the light dose stimulation in tomatoes and the requirement for day length [44]. SELF-PRUNING (*sp*) mutations in tomato resulted in single amino acid changes, and the mutant phenotype was simulated by the overexpression of SP antisense RNA. The ectopic and overexpression of SP and CEN transgenes in tomato lead to uncertain phenotypes, regulating leaf substitution for flowers in inflorescences and inducing the vegetative apex to become a reproductive branch. SP genes are expressed in bud tips and leaves from early stages and later also in the inflorescence and flower primordium. This expression pattern was similar to those of the tomato direct homologs LEAFY and FLORICAULA [45]. The SP interaction protein of tomatoes defines a conservative signaling system that regulates bud structure and flowering as well as the vegetative to reproductive transition of the zygote meristem [46]. The tomato blind gene encodes an MYB transcription factor that controls lateral meristem formation [47]. In some previous studies on tomato, some genes control the growth of tomato plants, and we can speculate that these genes may have a close effect on tomato inflorescence branches.

Changes in inflorescence branches determine the number of flowers and thus reproductive success and crop yield. In previous studies, it has been shown that AN controls inflorescence structure by promoting the continuous stage of inflorescence meristem development toward flower specification. During this gradual phase transition, the loss of ANANTHA (AN) genes delays flower formation, leading to additional branching. The variation in Solanaceae inflorescences is regulated by the time changes acquired by flower fate, providing a flexible evolutionary mechanism for the development of synctic inflorescence buds [4]. The common expression of homologs of AN in petunias results in a very early flowering and conversion of inflorescences to single flowers and leaves to petals. The ectopic expression of AN activates the identity of organ primordia or genes required for growth. Previous studies have shown that AN plays a broader role in flower pattern formation than previously thought and that the different roles of AN homologs in the spatiotemporal control of flower identity in different species are caused by their different expression patterns [48]. FALSIFLORA (FA) mutants are not able to form complete flowers, and FA is also able to control flowering time and floral meristem development [20]. The *fa* mutation primarily alters inflorescence development, causing flowers to be replaced by secondary branches but also producing a late flowering phenotype, with an increase in the number of leaves below the first and successive inflorescences. The model suggested that FA loci regulated floral meristem identity and flowering time in tomato in a similar way to the role of FLORICAULA (FLO) and LEAFY (LFY) genes in *A**rabidopsis thaliana*. The gene is expressed in the tomato plants meristem and flower meristem, leaf primordium and leaf, and four flower organs. Compared with other FLO/LFY lineal homologs, the function of this gene was analyzed in tomato, a plant with a synphytic growth habit and cyme development. As the *fa* mutant plant cannot form complete flowers, it produces four rounds of flowers, albeit modified, which can be used to study the functional link between flower induction and floral organ specification [49]. The results of our study are highly consistent with those of previous studies, indicating that the different materials of the branches of the two inflorescences may be influenced by mutations of AN (*an*) and FA (*fa*) genes.

## 4. Materials and Methods

### 4.1. Test Materials

Compound inflorescences (represented by CI in the transcriptomic analysis, where CI corresponds to 115 in the raw sequencing data) and single racemes (identified as SR in the transcriptomic analysis, where SR corresponds to 116 in the raw sequencing data) were used as parents in this study, in which the female had compound inflorescences (CIs) and the male had single racemes (SRs) (both provided by the Tomato Research Group of Northeast Agricultural University) (Figure 9). In a greenhouse at the Horticulture Station of Northeast Agricultural University (Harbin, China), tomato seeds are sown in pots filled with soil and grown under controlled conditions (light 16 h, 25 °C, and ambient humidity 50%). All plants were kept at 25 degrees Celsius and 50% relative humidity. In the early, middle, and late stages of inflorescence differentiation, the parental flower buds were taken once, and 3 biological replicates were prepared for each treatment.

### 4.2. Paraffin Sectioning

To observe the development of the tomato inflorescence meristem, we examined paraffin sections of tomato inflorescence tissue in early and middle stages [50]. The definition of three stages of tomato inflorescence: the early inflorescence of tomato was defined as about 3 days after germination; the middle inflorescence of tomato was defined as about 10 days after inflorescence growth; the late inflorescence of tomato was defined as about 15 days. The slices were placed in xylene for 40 min, anhydrous ethanol for 10 min, and 75% alcohol for 5 min and then rinsed with tap water. The slices were immersed in saffron dye and stained for 1–2 h with tap water, and the excess dye was washed away. The samples were sliced and subjected to a 50%, 70%, and 80% gradient alcohol decolorization. The slices were dyed with solid green dye for 30–60 s, soaked in three cylinders of anhydrous ethanol, and dehydrated. The slices were divided into n-butanol and xylene transparent sections for 5 min each. The slices were removed from the xylene and slightly dried. Neutral gum was used to seal the slices for microscopic examination, and the images were collected and analyzed.

### 4.3. RNA Extraction and Illumina Sequencing

In this experiment, we sequenced the inflorescence transcriptomes of two different tomatoes using the DNA Nano Ball Sequence (DNBSEQ) platform. A total of 18 samples were analyzed, and each sample yielded an average of 6.72 G of data. Total RNA was extracted from leaf samples using plant RNA Mini-kits according to the manufacturer’s instructions (Watson, China). Using the RNAprpe Pure Plant Kit (Thermo Fisher, New York, NY, USA), total RNA was extracted from a total of 18 samples in each group for real-time quantitative PCR (qRT-PCR) and RNA sequencing (RNA-seq) analyses. The extracted total RNA was delivered to BGI (Shenzhen, China) for high-throughput sequencing. The company provides the following detailed experimental methods. mRNA samples were isolated from total RNA samples treated with DNA enzyme I by the oligo (dT) method. Eighteen cDNA libraries were constructed from purified RNA. The library was then sequenced using Illumina HiSeq4000 machines according to the Illumina protocol [51,52].

### 4.4. DEG Identification

Gene expression levels were quantified by standardizing read counts to aligned fragments (FPKM) of transcripts per kilobase of reads per million markers [53,54]. The fractional index of transcription splicing alignment (HISAT) was used to align pairs of clean reads with the reference genome [54,55]. We used NOIseq in conjunction with noise distribution models to detect DEGs [56,57]. Through the comparison of gene expression among varieties, some DEGs were screened for further enrichment analysis. The read count data representing the number of reads contained in the transcript were used as the input data in the analysis of differential gene expression. Through the comparison of gene expression among varieties, some DEGs were screened for further enrichment analysis. The read count data representing the number of reads contained in the transcript were used as the input data in the analysis of differential gene expression. DEGs were defined as genes with a log_2_FC change ≥ 1 and FDR value ≤ 0.05 [58,59]. Multiple hypothesis testing was used to correct the error, and the false detection rate (FDR) was used to control the threshold of the *p* value to avoid false positives.

### 4.5. KEGG Analysis of Differentially Expressed Genes

KEGG (Kyoto Encyclopedia Genes and Genomes) database enrichment analysis was performed using the Phyper function in R software [57]. The X-axis is the ratio of the number of genes annotated to an item in the selected gene set to the total number of genes annotated to the item in the species. The calculation formula is as follows: Rich Ratio = Term Candidate Gene Num/Term Gene Num, and the Y axis is the KEGG pathway: the bubble size represents the number of genes annotated to a certain KEGG pathway, and the color represents the enrichment significance value. FDR (false discovery rate) correction was performed for the *p*-value. Generally, a Q-value ≤ 0.01 indicated significant enrichment. The redder the color is, the smaller the significance value.

### 4.6. GO Enrichment Analysis of Differentially Expressed Genes

This software used in the experiments for the GOseq (Young et al. 2010) Gene Ontology ((GO), http://www.geneontology.org/) [60] GO functional significance enrichment analysis provided the GO functional items (terms) that are significantly enriched in candidate genes compared with the whole genetic background of this species, thus revealing the biological functions that candidate genes are significantly associated with. The analysis first put all candidate genes into the Gene Ontology database (http://www.geneontology.org/) of each map entry, calculated each item on the number of genes, and then, using hypergeometric inspection, found out that, compared with all the background of this species gene, GO was significantly enriched in candidate genes entries. We used R basis function phyper for *p* value calculation (https://stat.ethz.ch/R-manual/R-devel/library/stats/html/Hypergeometric.html (accessed on 10 June 2022)). The multiple test was positive, then on the *p* value, the correction software package was the *p* value (https://bioconductor.org/packages/release/bioc/html/qvalue.html (accessed on 10 June 2022)). Ending with a Q value (corrected *p* value) ≤0.05 was the threshold, and the GO term meeting this condition was defined as the GO term significantly enriched in candidate genes.

### 4.7. Weighted Coexpression Gene Network Analysis (WGCNA)

In this study, 18 transcriptome samples (2 materials at 3 time points, each repeated 3 times) were selected for WGCNA analysis. The WGCNA software package was used for gene co-expression network analysis; the X-axis represents phenotypes (samples) and the Y-axis represents different co-expression gene modules [61,62]. For missing values, if the gene expression of a sample was missing more than 50%, it was eliminated; if a gene was missing in more than 50% of the samples, it was eliminated. For the filtering of gene expression, the top 75% of the genes with the median absolute deviation (MAD) were retained. The filtered data used the automatic network construction function blockwiseModules to construct a co-expression network. The filtered data constructed a co-expression network through the automatic network construction function blockwiseModules. The network type was unsigned, and the correlation type was Pearson. In the module, the exceptions were that the power was 9, the TOM similarity threshold was 0.19, and the minimum number of genes was 20. To study highly correlated modules, Pearson correlation coefficients between sample matrices and gene modules were calculated and statistically tested. The higher the correlation coefficient, the higher the correlation between the module and the sample.

### 4.8. Related Genes Were Verified by qRT–PCR

Real-time quantitative PCR was used to test the reliability of the RNA-seq data.

In this experiment, 10 DEGs were randomly selected. Primers were designed based on the NCBI database, and reverse transcription and fluorescence measurements were performed according to the requirements of the kit (Vazyme, Nanjing, China). These were mainly involved in plant hormone signal transduction and metabolic pathways. Appendix A shows information regarding the design of qRT–PCR primers. Ace Q^®^q PCR SYBR^®^Green Master Mix (Vazyme, NJ, USA,) and the qTOWER3G detection system (Analytik Jena, Germany) were used in this study [63]. Gene expression was analyzed by the 2^−∆∆CT^ method [64].

## 5. Conclusions

To study the cause of different inflorescence branches in tomato, we used two different inflorescence materials for transcriptome analysis. In the early, middle, and late comparison groups of tomato inflorescences, a large number of DEGs were found. According to GO and KEGG enrichment analyses, DEGs were mainly enriched in metabolic processes, cellular processes, and biological regulation pathways. Based on previous studies, we screened 41 genes that we believed might be related to tomato inflorescence branching from the early and late comparison groups. To better analyze the functions of the DEGs, we used MapMan software to interpret and classify DEGs and concluded that biological regulation, plant hormone signal transduction, and metabolic regulation pathways all have certain effects on tomato growth and development and inflorescence branching.

## Figures and Tables

**Figure 1 ijms-23-08216-f001:**
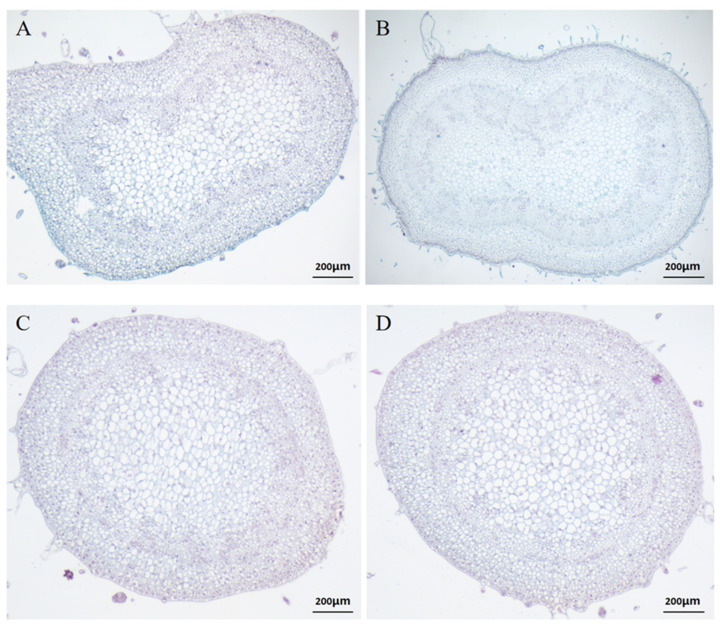
Images of paraffin sections of tomato inflorescences. (**A**): represents E_CI, and E_CI represents the early stage of the compound inflorescence. (**B**): represents M_CI, which represents the middle stage of the compound inflorescence. (**C**): represents E_SR, and E_SR represents the early stage of a single raceme. (**D**): represents M_SR, which represents the metaphase of a single raceme.

**Figure 2 ijms-23-08216-f002:**
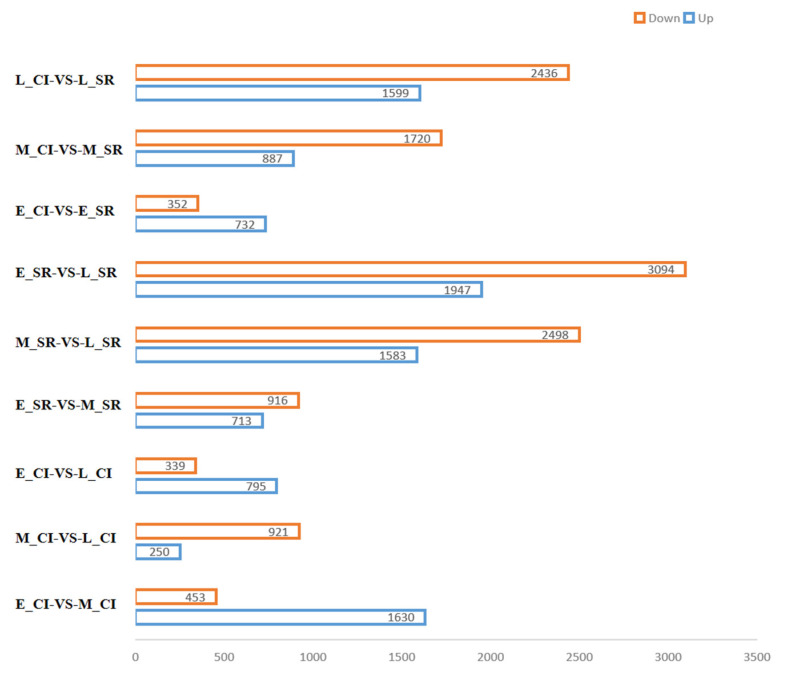
DEGs identified from different comparisons.

**Figure 3 ijms-23-08216-f003:**
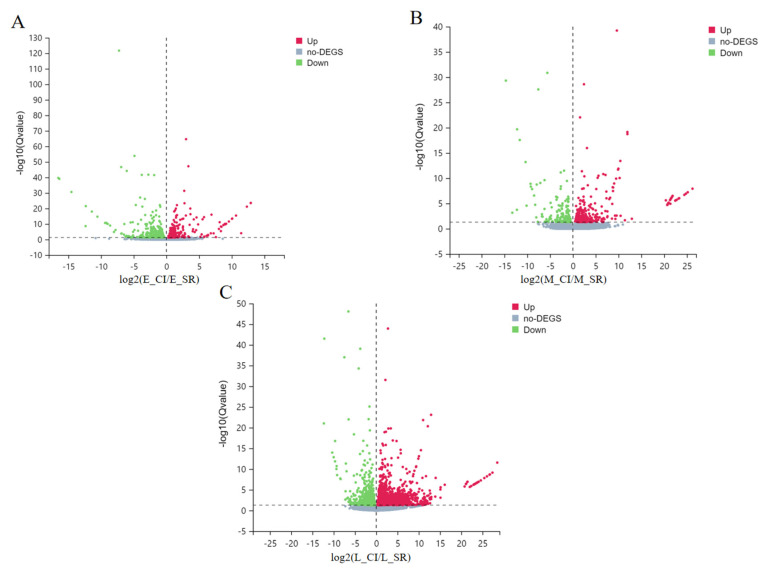
DEG volcano map. (**A**) E116-vs.-E115 volcano map. (**B**) M116-vs.-M115 volcano map. (**C**) L116-vs.-L115 volcano map. Red dots represent up-regulated genes, green dots represent down-regulated genes, and gray dots represent non-DEGs. The X-axis represents the fold change of the difference after conversion to log2 values, while the Y-axis represents the significance value after conversion to −log10 values.

**Figure 4 ijms-23-08216-f004:**
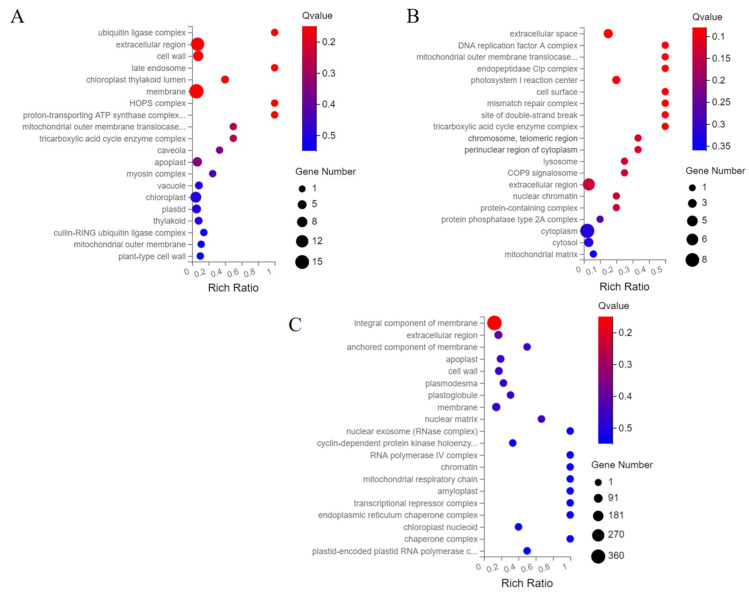
GO functional enrichment of differentially expressed genes in tomato inflorescences. (**A**): E_SR-vs.-E_CI functional enrichment bubble diagram. (**B**): M_SR-vs.-M_CI functional enrichment bubble diagram. (**C**): L_SR-vs.-L_CI functional enrichment bubble diagram. The size of the bubble indicates the number of genes enriched in the GO term. The color of the bubbles indicates the Q value. The enrichment factor is the ratio of the number of differentially expressed genes annotated in the pathway to the number of all genes annotated in the pathway. The higher the enrichment factor is, the higher the enrichment degree of the pathway.

**Figure 5 ijms-23-08216-f005:**
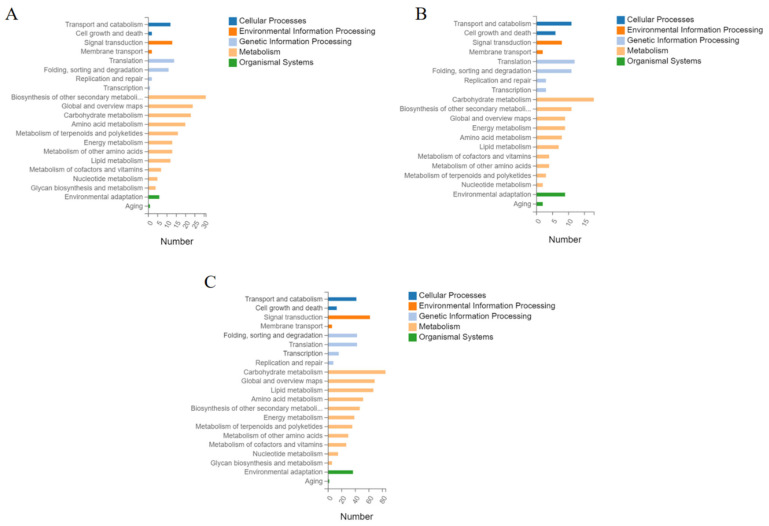
KEGG pathway annotation classification of differentially expressed genes. (**A**) represents the E_SR-vs.-E_CI pathway annotation classification. (**B**) represents the M_SR-vs.-M_CI KEGG pathway annotation classification. (**C**) represents the L_SR-vs.-L_CI KEGG pathway annotation classification. The X-axis represents annotation to a certain KEGG number of genes in the pathway category, and the Y-axis is the KEGG pathway category.

**Figure 6 ijms-23-08216-f006:**
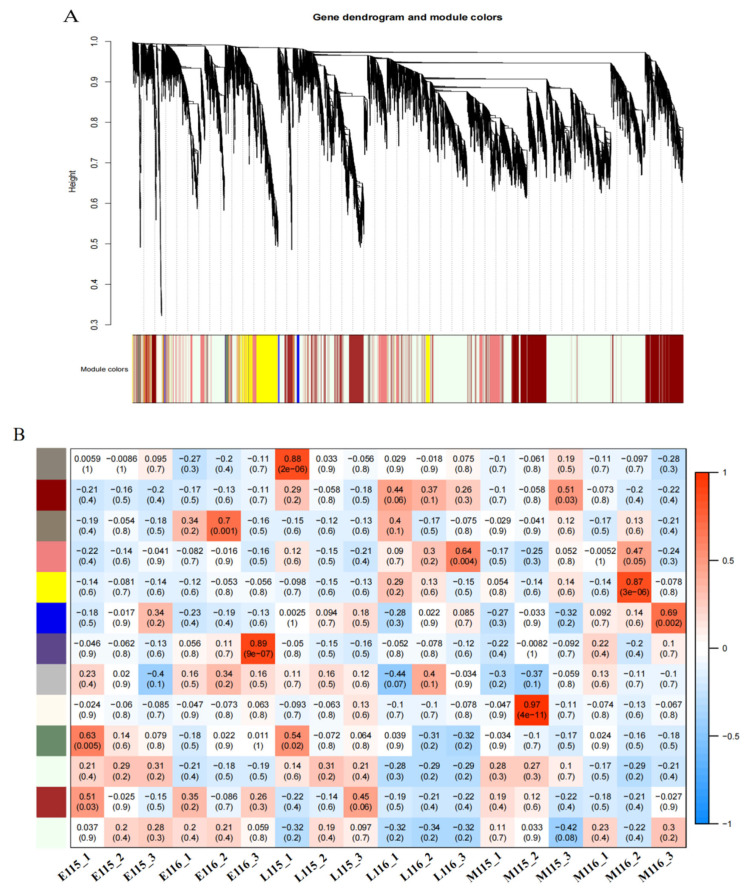
Gene co-expression network analysis by WGCNA. (**A**) Gene clustering tree and module division. (**B**) Module-sample association. The abscissa represents the samples; the ordinate represents the modules. The variation from blue (low) to red (high) indicates the ranges of the DEGs. (**C**) Enrichment in KEGG pathways of DEGs in the dark red module. (**D**) Enrichment in KEGG pathways of DEGs in the yellow module.

**Figure 7 ijms-23-08216-f007:**
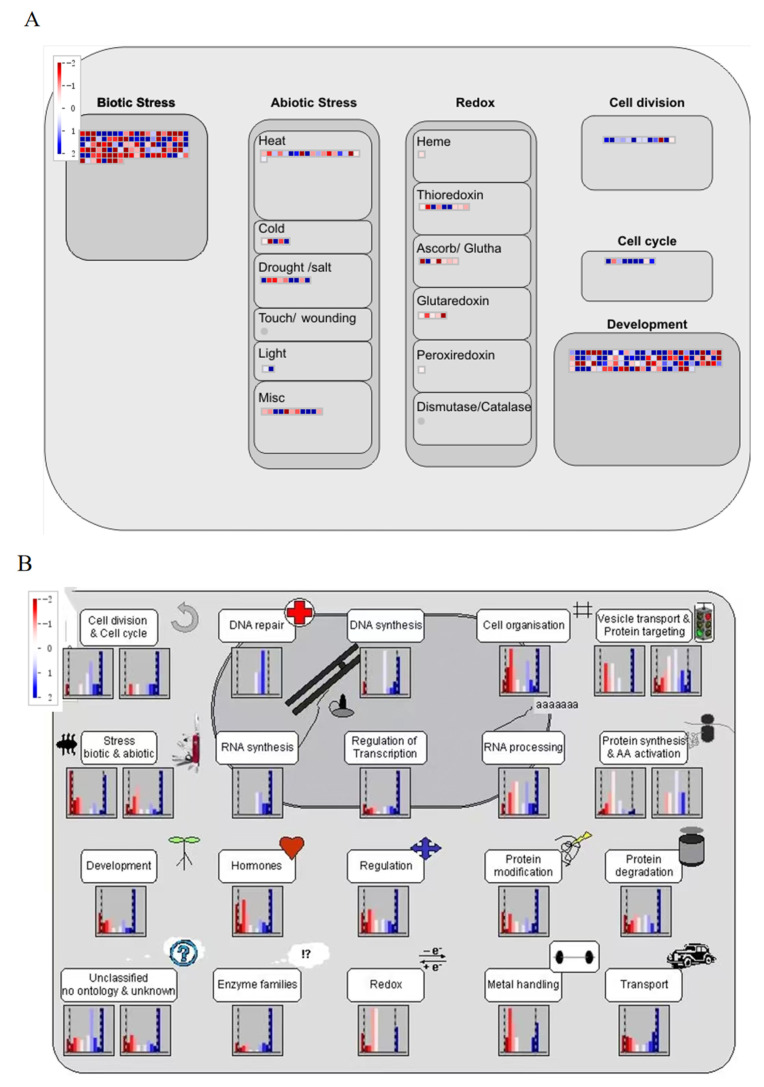
Clustering patterns of DEGs generated using the MapMan tool. (**A**) Cellular response of DEGs in L_SR-vs.-L_CI. (**B**) Cell functions of DEGs in L_SR-vs.-L_CI. (**C**) Biological regulation of DEGs in L116 vs. L115. (**D**) Metabolic regulation of DEGs in L_SR-vs.-L_CI.

**Figure 8 ijms-23-08216-f008:**
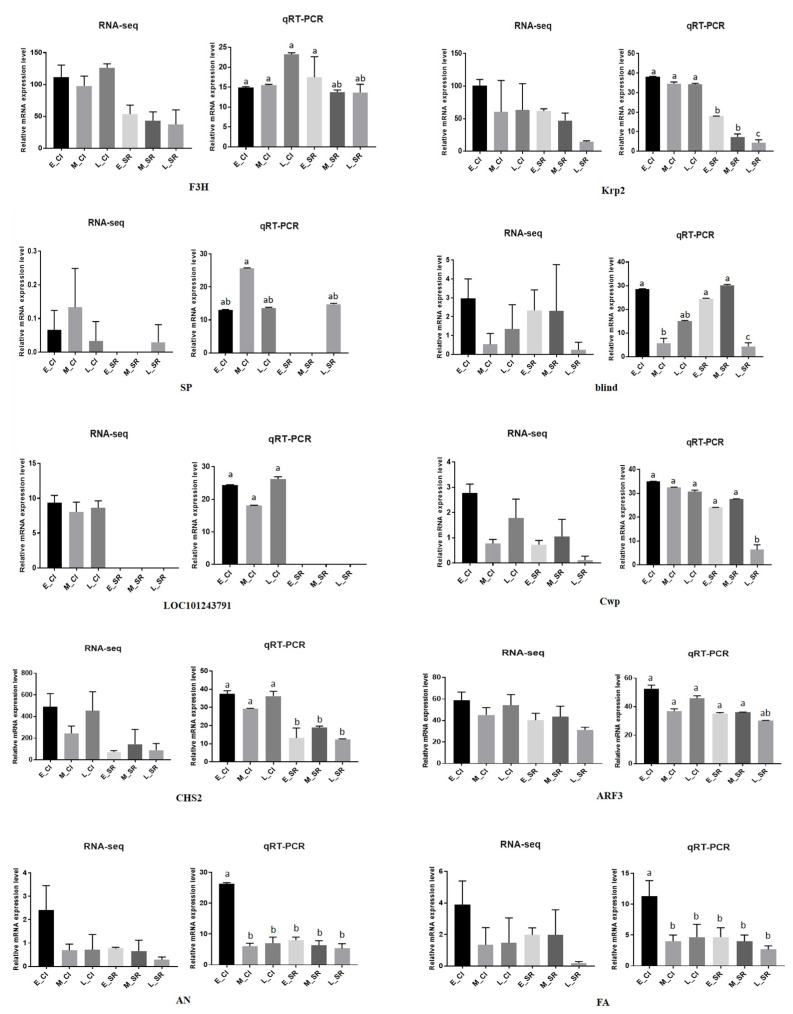
Comparative analysis of expression results between RNA-Seq and qRT–PCR for 10 DEGs. Three technical replicates were performed for each biological replicate of each sample. Error bars represent standard deviation. *p* < 0.05 means significant difference: a was significantly higher than b and c; b was significantly higher than c.

**Figure 9 ijms-23-08216-f009:**
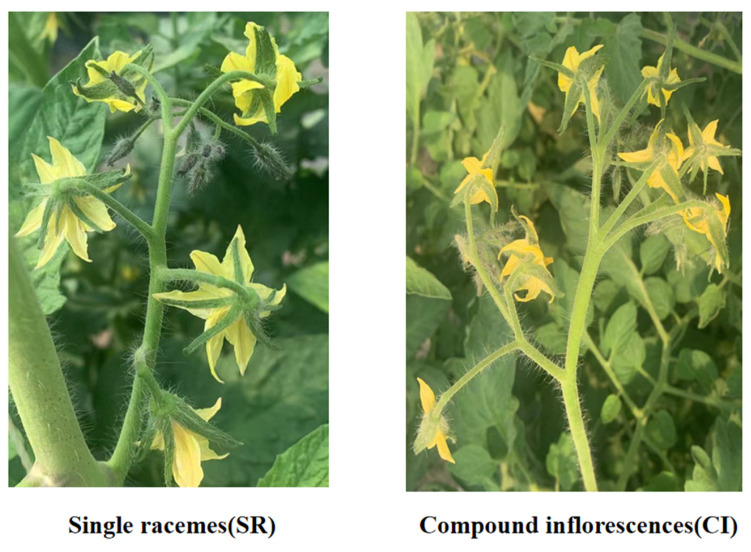
Appearance of the parental materials used for transcriptome analysis. SR was used as the male parent, and CI was used as the female parent.

**Table 1 ijms-23-08216-t001:** Statistics of the tomato transcriptome database.

Sample	Total Mapping (%)	Uniquely Mapping (%)	Total Raw Reads (M)	Total Clean Reads (M)	Total Clean Bases (Gb)	Clean Reads Q20 (%)	Clean Reads Q30 (%)	Clean Reads Ratio (%)
E_CI_1	92.54	91.01	47.33	45.03	6.76	96.58	91.56	95.15
E_CI_2	92.9	91.47	47.33	45.18	6.78	96.62	91.70	95.47
E_CI_3	93.23	91.85	45.57	43.92	6.59	96.71	91.86	96.37
E_SR_1	92.72	91.29	47.33	45.40	6.81	96.81	92.13	95.92
E_SR_2	92.92	91.44	47.33	45.33	6.80	96.59	91.55	95.77
E_SR_3	92.24	90.81	47.33	44.78	6.72	96.64	91.67	94.63
M_CI_1	93.10	91.68	45.57	44.16	6.62	96.66	91.72	96.89
M_CI_2	93.11	91.66	45.57	44.12	6.62	96.58	91.56	96.81
M_CI_3	92.85	91.38	47.33	45.39	6.81	96.86	92.20	95.91
M_SR_1	93.04	91.56	45.57	43.84	6.58	96.66	91.68	96.20
M_SR_2	92.70	91.28	47.33	45.08	6.76	96.76	91.93	95.24
M_SR_3	92.98	91.55	47.33	45.46	6.82	96.59	91.55	96.06
L_CI_1	92.64	91.19	47.33	45.43	6.81	96.67	91.79	95.99
L_CI_2	92.49	91.10	47.33	45.01	6.75	96.72	91.92	95.11
L_CI_3	93.06	91.59	47.33	45.42	6.81	96.73	91.88	95.98
L_SR_1	93.37	91.92	45.57	43.86	6.58	96.69	91.73	96.25
L_SR_2	92.91	91.44	47.33	45.44	6.82	96.48	91.31	96.01
L_SR_3	93.44	91.96	45.57	43.99	6.60	96.65	91.69	96.52

E_CI indicates early compound inflorescence development; E_CI_1, E_CI_2, and E_CI_3 are three samples from the early stage of compound inflorescence development in E_CI. M_CI represents the middle of compound inflorescence development, and M_CI_1, M_CI_2, and M_CI_3 are three samples from the middle stage of compound inflorescence development in M_CI. L_CI indicates late compound inflorescence development, and L_CI_1, L_CI_2, and L_CI_3 are three samples from the late compound inflorescence of L_CI. E_SR indicates early single raceme development, and E_SR_1, E_SR_2, and E_SR_3 are three samples of a single raceme. M_SR indicates the middle of single raceme development, and M_SR_1, M_SR_2, and M_SR_3 are three samples from the middle stage of a single raceme. L_SR indicates late single raceme development, and L_SR_1, L_SR_2, and L_SR_3 are three samples from the late stage of a single raceme.

**Table 2 ijms-23-08216-t002:** Affect-related genes of tomato inflorescence traits.

Gene ID	Symbol	Log_2_ Fold-Chang		Pathway
		E116-vs-E115	L116-vs-L115	
101252689	LOC101252689	0.98	1.16	Plant hormone signal transduction
101267446	LOC101267446	0.96	1.12	Plant hormone signal transduction
778363	ARF3	0.5	0.64	Plant hormone signal transduction
101262709	LOC101262709	1.38	1.85	Plant hormone signal transduction
778241	ARF4	0.51	1.11	Plant hormone signal transduction
101248511	LOC101248511	−1	−0.93	Plant hormone signal transduction
100134907	GA2ox1	−2.59	−3.24	Plant hormone signal transduction
101258926	LOC101258926	−3.04	−2.2	Plant hormone signal transduction
100736482	F3H	1.01	1.63	Flavonoid biosynthesis
101249699	LOC101249699	1.32	1.44	Flavonoid biosynthesis
101266223	LOC101266223	0.95	1.24	Flavonoid biosynthesis
104648079	LOC104648079	0.93	1.24	Zeatin biosynthesis
101256765	LOC101256765	−4.5	−4.19	Zeatin biosynthesis
101243791	LOC101243791	10.07	9.9	Carbon metabolism
100529127	LOC100529127	4.85	7.92	Cell cycle
543785	krp2	0.66	1.93	Cell cycle
101254782	LOC101254782	1.58	1.14	Chloroplastic
101255133	LOC101255133	1.47	1.57	Chloroplastic
101255274	LOC101255274	8.92	8.65	Chloroplastic
101258716	LOC101258716	2.11	1.91	Chloroplastic
101261442	LOC101261442	1.12	1.16	Chloroplastic
101266861	LOC101266861	1.08	1.56	Chloroplastic
104649068	LOC104649068	1.26	1.3	F-box
112940016	LOC112940016	6.79	6.53	F-box
101250115	LOC101250115	12.31	12.13	F-box
101244792	LOC101244792	9.56	9.4	F-box family protein
101261682	LOC101261682	3.65	5.77	F-box protein CPR1
778295	CHS2	2.73	2.18	Circadian rhythm
101246029	SFT	−1.3	−4.15	Circadian rhythm
100301925	AN	1.59	1.12	Making of a compound inflorescence
543630	FA	0.94	2.61	Floral meristem identity
544038	SP	1.86	1.8	Regulates vegetative to reproductive switching
543703	blind	0.29	2.33	Control the formation of lateral meristems
101259221	LOC101259221	0.79	1.11	Starch and sucrose metabolism
101266123	LOC101266123	0.44	1.31	Glycine, serine and threonine metabolism
101268249	LOC101268249	0.52	0.54	Purine metabolism
101268552	LOC101268552	5.39	5.13	Sulfur metabolism
112940464	LOC112940464	8.95	8.75	Fructose and mannose metabolism
101245153	CYP736A1	1.35	1.83	Tissue specific promoters
101251053	dxs2	0.8	2.15	Terpenoid backbone biosynthesis
101261618	Cwp	1.9	3.77	Cuticular water permeability protein

## Data Availability

The raw sequencing data of this article are stored in the NCBI Sequence Read Archive under accession number PRJNA847367.

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
