# Peer review of "Transcriptome Analysis to Explore the Cause of the Formation of Different Inflorescences in Tomato"

_ijms, 2022, doi:10.3390/ijms23158216_

Round 1
Reviewer 1 Report
The resubmitted version has a major improvement in the introduction and the methods compared to previous versions, which makes the manuscript easier to read. There are still some mistakes that the authors want to pay attention:
1. I noticed that the figure legend for Figure 5 is deleted but the figure is still in there.
2. For the WGCNA analysis, the authors mentioned performing KEGG enrichment on dark red and yellow modules in text (line 258-259), but the figure legend for Figures 6C and 6D, stated brown and green modules. Could you clarify this? What exactly modules did you choose to perform KEGG analysis and why did you choose them?
3. I saw many typos, please take a read-through and correct your typos.
Author Response
Point 1. I noticed that the figure legend for Figure 5 is deleted but the figure is still in there.
Response 1. Thank you very much for your responsible comments. The editor's opinion asked me to mark the changes, so I deleted Figure 5 according to your opinion, but Figure 5 will still be displayed. The display is for a better view of the changes. You have already seen the changes in Figure 5, so it has been deleted in the article. Thank you for your advice.
Point2. For the WGCNA analysis, the authors mentioned performing KEGG enrichment on dark red and yellow modules in text (line 258-259), but the figure legend for Figures 6C and 6D, stated brown and green modules. Could you clarify this? What exactly modules did you choose to perform KEGG analysis and why did you choose them?
Response 2.Thank you very much for your kind advice. It is correct to mention KEGG enrichment for dark red and yellow modules. Sorry for 6C and 6D are writing errors. 6C is dark red module enrichment, 6D is yellow module enrichment. In addition, I have changed 6C to dark red module enrichment and 6D to yellow module enrichment in lines 265-266 of this article.
These two modules were chosen because there are more co-expressed genes enriched in inflorescence-related pathways in these two modules. Let us see more clearly, inflorescence branch related genes mainly enriched pathways. Therefore, I focus on these two modules, as hinted at in lines 252-257 of this article.
Point 3. I saw many typos, please take a read-through and correct your typos.
Response 3. Thank you very much for your kindly advice. I will read the article carefully and correct the misspellings.
Finally, the manuscript has been revised according to comments of reviewers and editor. All changes in the manuscript were marked by red color. If there is any shortage, please inform us! We will try our best to revise this manuscript.
Thank you very much for your attention and kind advice.
Reviewer 2 Report
The authors propose to set out to expire the transcriptomic differences between distinct types if tomato branching types (cymes vs racymes).
The idea seems interesting as it could benefit producers by identifying more easily variants with higher agronomic value.
Saying that, I find that although the results derived from the transcriptomic pipeline are sound I can not understand how the authors chose their biological material. There is an overall lack of explanation in that domain and I simply can not relate the genetic expression data with stages of development that are on the most part unclear. e.g. the paraffin slides are from inflorescence meristems or early flower meristems? The idea of early and late needs to be clarified. Then there is the idea of the parental types, maternal and paternal that needs to be clarified (hermaphrodite? dioecious? monoecious?).
There are some sentences throughout the text that need to be revised or better explained:
line 27 - favourable branching
line 48 - co-flowering growth
line 30 - SIM, should be shoot inflorescence meristem?
line 96 - explain known inflorescence branching sites. This is quite important as it might justify why the authors chose those specific developmental stages
line 106 - can't understand the meaning of the sentence
line 112 - the definition of early and late needs to be supported by additional morphological data
Author Response
Point 1. Saying that, I find that although the results derived from the transcriptomic pipeline are sound I can not understand how the authors chose their biological material. There is an overall lack of explanation in that domain and I simply can not relate the genetic expression data with stages of development that are on the most part unclear. e.g. the paraffin slides are from inflorescence meristems or early flower meristems? The idea of early and late needs to be clarified. Then there is the idea of the parental types, maternal and paternal that needs to be clarified (hermaphrodite? dioecious? monoecious?).
Response 1. Thank you very much for your kind advice. The biomaterials in this paper were obtained by recombining inbred lines.
In the paraffin section, Figure 1A and Figure 1C are the early inflorescence, and Figure 1B and Figure 1D are the middle inflorescence. The definition of three stages of tomato inflorescence: the early inflorescence of tomato was defined as about 3 days after germination; The middle inflorescence of tomato was defined as about 10 days after inflorescence growth. The late inflorescence of tomato is defined at about 15 days. Lines 459-463 in the materials method are also modified and explained.
The tomato is hermaphrodite plants, tomato is plant belonging to the nightshade family, nightshade plants are bisexual flowers, that is, monoecious plants with the same flowers.
Point 2. line 27 - favourable branching
Response 2. Thank you very much for your responsible comments. Plant yield is related to the number of branches of inflorescence, the more branches, the higher the yield without affecting its quality. Favourable branching refers to the number of branches without affecting the optimal production high.
Point 3. line 48 - co-flowering growth
Response 3. Thank you very much for your kind advice. Sorry this is a clerical error. I have modified “co-flowering growth” to “sympodial growth” in line 48 of the article.
Point 4. line 30 - SIM, should be shoot inflorescence meristem?
Response 4. Thank you very much for your valuable advice. SIM is short for sympodial inflorescence meristem. It was amended in line 30 of the article.
Point 5. line 96 - explain known inflorescence branching sites. This is quite important as it might justify why the authors chose those specific developmental stages
Response 5. Thank you very much for your kind advice. I'm sorry, I don't quite understand your question. Can I understand you asking about the gene that controls the inflorescence branching. Genes controlling inflorescence branches FAB, FIN, SFT, J, FA, TMF, AN, S, J2, and EJ2 have been found in tomato and are also mentioned in lines 42-70 of the introduction. Most inflorescence branches change because these genes are mutated during plant growth and development. In order to understand the reasons for controlling tomato inflorescence branching in this paper, we conducted data analysis on the early, middle and late periods of the material and observed the gene changes in these three periods. To find out the causes of inflorescence branching.
Point 6. line 106 - can't understand the meaning of the sentence
Response 6. Thank you very much for your comments. This “analyzed whether the compound inflorescence of tomato was regulated by the same genes as the compound inflorescence of tomato studied by predecessors through a large amount of data” means “the causes of different inflorescence branches of tomato inflorescence were analyzed through a large amount of data and the similarities and differences of genes controlling inflorescence branches were observed.” I'm sorry that I didn't understand what I wrote. I have modified it in line 106.
Point 7. line 112 - the definition of early and late needs to be supported by additional morphological data
Response 7. Thank you very much for your valuable advice. The definition of the three stages of inflorescence is explained in line 459-463 of the material method in this paper (The definition of three stages of tomato inflorescence: the early inflorescence of tomato was defined as about 3 days after germination; The middle inflorescence of tomato was defined as about 10 days after inflorescence growth. The late inflorescence of tomato is defined at about 15 days). I am very sorry for not providing a more intuitive picture display.
Finally, the manuscript has been revised according to comments of reviewers and editor. All changes in the manuscript were marked by red color. If there is any shortage, please inform us! We will try our best to revise this manuscript.
Thank you very much for your attention and kind advice.
Reviewer 3 Report
The paper presented for review concerns transcriptomic analyzes of tomatoes in order to investigate the formation of two different inflorescences. Although it is based only on transcriptomic research, it presents interesting and worth publishing results. Subject to corrections, it could be published in IJMS.
1) The abstract is very badly written. It should definitely not include an explanation of the abbreviations, such as DEGs, GO, or KEGG. The sentence "This study not only provides a theoretical basis for understanding inflorescence branching but also new ideas for inflorescence research." should be redrafted. Overall, I would suggest rewriting the summary. Briefly present the subject of experiments and what has been tested, and then present the most interesting results and conclusions.
2) In the materials and methods, please include a table or a diagram presenting individual experimental variants. This will allow the reader to quickly understand the experimental design.
3) In my opinion, 3 repetitions in the transcriptomic analysis is very little. This does not allow for a correct statistical analysis of the results. But we won't change that anymore.
4) Line 471 - although this is the beginning of the sentence, I suggest writing mRNA.
5) Subchapter 1.3 - Materials and methods - Please write what kit was used to construct the sequencing libraries and sequencing (kit names).
6) Line 550 - what were the primers designed on the basis of? Sequences obtained from sequencing or sequences from NCBI ?
7) Lines 420 - Arabidopsis with a capital letter
8) We write all names of Arabidopsis thaliana in italics (example lines 29, 41)
Author Response
Point 1. The abstract is very badly written. It should definitely not include an explanation of the abbreviations, such as DEGs, GO, or KEGG. The sentence "This study not only provides a theoretical basis for understanding inflorescence branching but also new ideas for inflorescence research." should be redrafted. Overall, I would suggest rewriting the summary. Briefly present the subject of experiments and what has been tested, and then present the most interesting results and conclusions.
Response 1. Thank you very much for your responsible comments. I've rewritten the abstract into this "The number of inflorescence branches is an important agronomic character of tomato. The meristem differentiation and development pattern of tomato inflorescence is complex and its regulation mechanism is very different from that of other model plants. Therefore, in order to explore the cause of tomato inflorescence branching, transcriptome analysis was conducted on two kinds of tomato inflorescences (single racemes and compound inflorescences). According to the transcriptome data analysis, there were many differentially expressed genes of tomato inflorescences at early, middle and late stages. Then, Gene Ontology and Kyoto Encyclopedia of Genes and Genomes enrichment of DEGs were performed. DEGs is mainly enriched in metabolic pathways, biohormone signaling and cell cycle pathways. According to previous studies, 41 genes related to inflorescence branch were screened from DEGs. This study not only provides a theoretical basis for understanding inflorescence branching, but also provides a new idea for the follow-up study of inflorescence. This study not only provides a theoretical basis for understanding inflorescence branching, but also new ideas for inflorescence research."
Point2. In the materials and methods, please include a table or a diagram presenting individual experimental variants. This will allow the reader to quickly understand the experimental design.
Response 2.Thank you very much for your kind advice. According to your requirements, I have added Figure 10 to the article to represent the experimental process for readers to read.
Figure 10. The design process is shown in the figure.
Point 3. In my opinion, 3 repetitions in the transcriptomic analysis is very little. This does not allow for a correct statistical analysis of the results. But we won't change that anymore.
Response 3. Thank you very much for your kindly advice. We only did 3 biological replicates for material and financial reasons. In the future experiments, we will do more biological repetition according to your requirements to make the results more accurate and valuable.
Point 4. Line 471 - although this is the beginning of the sentence, I suggest writing mRNA.
Response 4. Thank you very much for your valuable advice. I changed “Gene” to “mRNA” as you requested. Due to material and funding problems, we only did 3 biological replicates. In future experiments, we will make more biological repetitions according to your requirements to ensure the stability and reliability of the results.
Point 5. Subchapter 1.3 - Materials and methods - Please write what kit was used to construct the sequencing libraries and sequencing (kit names).
Response 5. Thank you very much for your kind advice. “Using the RNAprpe Pure Plant Kit (Thermo Fisher, New York, NC USA), total RNA was extracted from a total of 18 samples in each group for real-time quantitative PCR (qRT-PCR) and RNA sequencing (RNA-seq) analyses” has been added in the material method according to your requirements.
Point 6. Line 550 - what were the primers designed on the basis of? Sequences obtained from sequencing or sequences from NCBI ?
Response 6. Thank you very much for your comments. The primer design is based on the sequence obtained by NCBI.
Point 7. Lines 420 - Arabidopsis with a capital letter.
Response 7. Thank you very much for your valuable advice. I'm sorry for my handwriting mistakes. I have changed line 420 “arabidopsis thaliana” to “Arabidopsis thaliana”.
Point 8. We write all names of Arabidopsis thaliana in italics (example lines 29, 41)
Response 8. Thank you very much for your valuable advice. I have changed all “Arabidopsis thaliana” in the article to “Arabidopsis thaliana”.
Finally, the manuscript has been revised according to comments of reviewers and editor. All changes in the manuscript were marked by red color. If there is any shortage, please inform us! We will try our best to revise this manuscript.
Thank you very much for your attention and kind advice.
Reviewer 4 Report
Manuscript "Transcriptome analysis to explore the cause of the formation of different inflorescences in tomato" by
authors Hui Ya Yang, Ting Ting Zhao, Yang Xiang Xu, Bin Jing Jiang, Fu Jing Li is a private botanical investigation into branching. It can be assumed that the vision of the issue proposed by the authors reflects the real state of affairs, it can be assumed that some of the patterns identified are random. The authors do not offer any essential concept or scheme of development of events during the formation of the hand. The use of such expensive methods with such an undersample can be applied and, despite the low probability of matching these results with data that can be obtained by subsequent researchers, may allow some assumptions. The disadvantage of this work is the lack of an interaction model in the form of a scheme of successive stages of raceme ontogenesis, ignoring differences in flowers, and the lack of information about the plant phenotype. So, in particular, some viruses are able to cause significant changes in the phenotype and flowers and brushes and shoots. The photo clearly shows the difference in the parameters of the flowers. It seems to me that the authors should link the phenotype and biochemical modulations both in the description and in the scheme. It is also required to clearly describe the future prospects of this rough experiment. And also to analyze in the introduction the reasons that cause such manifestations.
Author Response
Point 1. authors Hui Ya Yang, Ting Ting Zhao, Yang Xiang Xu, Bin Jing Jiang, Fu Jing Li is a private botanical investigation into branching. It can be assumed that the vision of the issue proposed by the authors reflects the real state of affairs, it can be assumed that some of the patterns identified are random. The authors do not offer any essential concept or scheme of development of events during the formation of the hand. The use of such expensive methods with such an undersample can be applied and, despite the low probability of matching these results with data that can be obtained by subsequent researchers, may allow some assumptions.
Response 1. Thank you very much for your kind advice. Based on your requirements and experimental results, I propose a bold hypothesis in lines 126-129 of the introduction. Assuming that “According to previous studies, 41 genes related to inflorescence branch were screened from DEGs. The presence of FALSIFLORA(FA) and ANANTHA(AN) genes among the 41 genes is the main reason for the different inflorescence branches in tomato”.
Point 2. The disadvantage of this work is the lack of an interaction model in the form of a scheme of successive stages of raceme ontogenesis, ignoring differences in flowers, and the lack of information about the plant phenotype. So, in particular, some viruses are able to cause significant changes in the phenotype and flowers and brushes and shoots.
Response 2. Thank you very much for your responsible comments. I understand your question about raceme individuals. Our material is obtained by recombining inbred lines and there will be no problem with the phenotype of the inflorescence. We've also given some serious thought to your concern about the virus causing changes in the inflorescence. However, our transcriptome materials are all grown in the same environment, and the environment is very suitable. In the selection of materials, is also extremely careful. The plants with selected materials grew well and were free from virus. I am sorry that I cannot provide the information of plant phenotype. I have added the photo of inflorescence phenotype in the material method, hoping it will be helpful to you. We will also add information about plant phenotypes in future experiments according to your opinions.
Point 3. The photo clearly shows the difference in the parameters of the flowers. It seems to me that the authors should link the phenotype and biochemical modulations both in the description and in the scheme. It is also required to clearly describe the future prospects of this rough experiment. And also to analyze in the introduction the reasons that cause such manifestations.
Response 3. Thank you very much for your kind advice. I am sorry that due to time constraints, I can only modify the link between phenotype and biochemical regulation in the introduction. In line 131-137 of the introduction, I also added the future prospect of the experiment and analyzed the reasons. Add the following “At the same time, plant hormones also affect the inflorescence branch. Gibberellin mainly affects tomato branches. We've been able to identify the main reasons for the different branches in tomatoes, both in terms of genes and hormones. It shows that transcriptome experiments can roughly control the formation process of compound inflorescence and lay a theoretical foundation for the subsequent study of compound inflorescence”.
Finally, the manuscript has been revised according to comments of reviewers and editor. All changes in the manuscript were marked by red color. If there is any shortage, please inform us! We will try our best to revise this manuscript.
Thank you very much for your attention and kind advice.
Round 2
Reviewer 3 Report
Thank you very much for all the answers and corrections. Unfortunately, I am not entirely satisfied.
Minor revision:
1) Abstract - has been slightly improved, but still not good. The last sentence that is repeated would need to be edited or discarded. It doesn't make sense. The most important information is also missing - the key results and summary. As for abbreviations such as DEG, GO or KEGG. They are commonly known and in the summary, they should appear instead of the complete names. More specifically, in the summary, we leave abbreviations. We explain them in the introduction.
2) New Figure 10 - please present the experimental variants and not necessarily the plan of the entire experiment.
3) Line 495 - convert MRNA to mRNA
4) 500 line - I did not mean such a change. Please change the mRNA to Gene
Author Response
Point 1. Abstract - has been slightly improved, but still not good. The last sentence that is repeated would need to be edited or discarded. It doesn't make sense. The most important information is also missing - the key results and summary. As for abbreviations such as DEG, GO or KEGG. They are commonly known and in the summary, they should appear instead of the complete names. More specifically, in the summary, we leave abbreviations. We explain them in the introduction.
Response 1. Thank you very much for your responsible comments. I've rewritten the abstract into this "The number of inflorescence branches is an important agronomic character of tomato. The meristem differentiation and development pattern of tomato inflorescence is complex and its regulation mechanism is very different from that of other model plants. Therefore, in order to explore the cause of tomato inflorescence branching, transcriptome analysis was conducted on two kinds of tomato inflorescences (single racemes and compound inflorescences). According to the transcriptome data analysis, there were many DEGs of tomato inflorescences at early, middle and late stages. Then, GO and KEGG enrichment of DEGs were performed. DEGs is mainly enriched in metabolic pathways, biohormone signaling and cell cycle pathways. DEGs were mainly enriched in metabolic pathways, and FALSIFLORA(FA) and ANANTHA(AN) genes were the most notable of 41 DEGs related to inflorescence branching. This study not only provides a theoretical basis for understanding inflorescence branching, but also provides a new idea for the follow-up study of inflorescence. "
Point2. New Figure 10 - please present the experimental variants and not necessarily the plan of the entire experiment.
Response 2.Thank you very much for your kind advice. I feel guilty that I failed to meet your expectation when I modified this problem last time. Figure 9 of the material method shows two experimental groups that are compared with each other. One is a raceme and the other a compound inflorescences. By comparing racemes and compound inflorescences, genes that regulate inflorescence branching are explored. I'm really sorry that I can't understand your meaning exactly. If I have a chance, I wonder if you could explain “the experimental variants” in more detail. I apologize for the problem.
Point 3. Line 495 - convert MRNA to mRNA
Response 3. Thank you very much for your kindly advice. I have converted MRNA to mRNA in line 495.
Point 4. 500 line - I did not mean such a change. Please change the mRNA to Gene
Response 4. Thank you very much for your valuable advice. I have changed mRNA to Gene in line 500.
Finally, the manuscript has been revised according to comments of reviewers and editor. All changes in the manuscript were marked by red color. If there is any shortage, please inform us! We will try our best to revise this manuscript. Thank you very much for your attention and kind advice.
This manuscript is a resubmission of an earlier submission. The following is a list of the peer review reports and author responses from that submission.
Round 1
Reviewer 1 Report
Transcriptome work is not able to find the cause, it is just an observation so I cannot agree with the authors that FA and AN are the causative genes. Genetic mutants have to be produced to discover the cause.
The following points the author may want to consider if wanted to resubmit to publication:
- Abstract needs to be improved - How many DEGs in total you have identified? What pathways are these DEGs involved in? What new ideas can be provided from this study?
- Introduction - It is not clear to me what is single racemes and what is compound inflorescences, these concepts need to be introduced. What is the hypothesis you want to test in this study? What is the main gaol to do this transcriptomic comparison?
- RNAseq libraries - I found your data is very similar to your other publication Yang et al., (2021). Are the CI from early, middle, and late stages from the same RNASeq in the other paper? If yes, you want to cite it, if not, please describe what cultivar you used in this study.
- RNAseq analysis - What statistical test do you use to identify DEGs? Do you do corrections to p-value? The significant DEGs have to be determined by the adjusted p-value after p-value corrections, otherwise, the DEGs you identified will have many false positives.
- GO analysis - Any reason you have two GO enrichment results (Figure 4 and Figure 5)? They seem redundant, and Figure 5 did not really give me useful information. The GO enrichment method needs to be improved. I am super confused about why you did DEG screening in GO enrichment analysis. What statistical model and tests did you use?
- WGCNA - why do you specifically look at the two modules, please clarified. What do you learn from the WGCNA analysis?
- MapMan analysis - What patterns do you want to present? I do not see any expression patterns in your results.
- Conclusion - The conclusion about FA and AN are the cause of inflorescence differentiation is invalid. The authors did a transcriptome analysis which is detecting gene expression levels. Gene mutations and causative effects are not able to detect by RNASeq. The authors did not even confirm the mutations.
Author Response
Point 1. Abstract needs to be improved - How many DEGs in total you have identified? What pathways are these DEGs involved in? What new ideas can be provided from this study?
Response 1. Thank you very much for your responsible comments.I have revised the abstract.
Point2. Introduction - It is not clear to me what is single racemes and what is compound inflorescences, these concepts need to be introduced. What is the hypothesis you want to test in this study? What is the main gaol to do this transcriptomic comparison?
Response 2.Thank you very much for your kind advice. I already described racemes and compound inflorescences in the last paragraph of the introduction. At the same time, the hypothesis and main purpose of verification are described
Point 3. RNAseq libraries - I found your data is very similar to your other publication Yang et al., (2021). Are the CI from early, middle, and late stages from the same RNASeq in the other paper? If yes, you want to cite it, if not, please describe what cultivar you used in this study.
Response 3. Thank you very much for your kindly advice. I use different materials than Yang et al., (2021) does. And CI is short for compound inflorescence. I have modified the materials used in this article in the materials method.
Point 4. RNAseq analysis - What statistical test do you use to identify DEGs? Do you do corrections to p-value? The significant DEGs have to be determined by the adjusted p-value after p-value corrections, otherwise, the DEGs you identified will have many false positives.
Response 4. Thank you very much for your recognition and encouragement. I used DESeq2 to identify and verify DEGs. And p will be corrected to ensure the effectiveness of DEGs.
Point 5. I GO analysis - Any reason you have two GO enrichment results (Figure 4 and Figure 5)? They seem redundant, and Figure 5 did not really give me useful information. The GO enrichment method needs to be improved. I am super confused about why you did DEG screening in GO enrichment analysis. What statistical model and tests did you use?
Response 5. Thank you very much for your valuable advice. We used bTU interactive platform for visibility analysis of data. The reason for the two results of GO enrichment is that Figure 4 mainly shows the main enrichment pathway of the differential genes in tomato inflorescence at different stages, while Figure 5 shows the classification of the differential genes in tomato inflorescence at different stages in three gene functions (molecular function, cellular process and biological process). Figures 4 and 5 represent different meanings respectively.
Point 6. WGCNA - why do you specifically look at the two modules, please clarified. What do you learn from the WGCNA analysis?
Response 6. Thank you very much for your comments. In these two modules, there are more coexpressed genes enriched in the inflorescence-related pathways. So I focused on analyzing these two modules. In particular, the co-expression network predicts the regulatory relationship between genes by using the expression correlation between genes.
Point 7. MapMan analysis - What patterns do you want to present? I do not see any expression patterns in your results.
Response 7. Thank you very much for your valuable advice. The analysis of MapMan has been modified.
Point 8. Conclusion - The conclusion about FA and AN are the cause of inflorescence differentiation is invalid. The authors did a transcriptome analysis which is detecting gene expression levels. Gene mutations and causative effects are not able to detect by RNASeq. The authors did not even confirm the mutations.
Response 8. Thank you very much for your comments. The conclusions have been revised.
Finally, the manuscript has been revised according to comments of reviewers and editor. All changes in the manuscript were marked by red color. If there is any shortage, please inform us! We will try our best to revise this manuscript.
Thank you very much for your attention and kind advice.
Reviewer 2 Report
The paper aims to uncover molecular determinants that sustain the development of morphologically distinct inflorescence branches, The subject is of interest as the type of inflorescence branches impact yield. Reading the abstract, the idea behind the study seems clear and the results interesting.
The introduction, however, requires extensive modifications as it is very fragmented. Some scientific terms lack care (e.g. lane 42) and some sentences are quite abstract and without clear meaning (e.g. lanes 31, 40, 57).
Results: Although the objective seems clear the biological material used in the study is not. The material and methods section descrition of the biological material is scarce and I can not understand what type of structures the authors are comparing (CI vs SR). A figure of the plants/inflorescences should be shown. It is also not clear what type of period the authors used to define "early middle or late". I also like to see longitudinal sections of the meristem at each type point the authors eventually choose.
Because I have questions about the biological material I found the subsequent transcritomic analysis hard to analyze, although, the pipeline to identify DEGs seems adequate.
Lines 324-359 of the discussion: How is the scientific content of the paragraph related to the results? If there is, it should be more clearly stated in the text.
Line 379 - snapurus?
line 380 - which plant meristem?
Author Response
Point 1. The introduction, however, requires extensive modifications as it is very fragmented. Some scientific terms lack care (e.g. lane 42) and some sentences are quite abstract and without clear meaning (e.g. lanes 31, 40, 57).
Response 1. Thank you very much for your kind advice. I have modified it in the introduction.
Point 2. Results: Although the objective seems clear the biological material used in the study is not. The material and methods section descrition of the biological material is scarce and I can not understand what type of structures the authors are comparing (CI vs SR). A figure of the plants/inflorescences should be shown. It is also not clear what type of period the authors used to define "early middle or late". I also like to see longitudinal sections of the meristem at each type point the authors eventually choose. Because I have questions about the biological material I found the subsequent transcritomic analysis hard to analyze, although, the pipeline to identify DEGs seems adequate.
Response 2. Thank you very much for your responsible comments. The racemes and compound inflorescences of tomato are shown as follows. Early inflorescence of tomato is defined as about 3 days after germinationï¼›The middle of a tomato inflorescence is defined as about 10 days after the inflorescence growsï¼›The late tomato inflorescence is defined as being about 15 days old.
Point 3. Lines 324-359 of the discussion: How is the scientific content of the paragraph related to the results? If there is, it should be more clearly stated in the text.
Response 3. Thank you very much for your kind advice. This section discusses the same differential genes in tomato racemes and tomato compound inflorescences at different stages. The pathway and function of each differential gene are described in detail in Table 2. Based on relevant literature, we review the role of differential genes in other plants and discuss genes that may have some influence on inflorescence development.
Point 4. ILine 379 - snapurus?
Response 4. Thank you very much for your valuable advice. I've changed the snapurus to arabidopsis thaliana.
Point 5. line 380 - which plant meristem?
Response 5. Thank you very much for your kind advice. It has been modified into tomato material in the article
Finally, the manuscript has been revised according to comments of reviewers and editor. All changes in the manuscript were marked by red color. If there is any shortage, please inform us! We will try our best to revise this manuscript.
Thank you very much for your attention and kind advice.
Round 2
Reviewer 1 Report
The authors did not address my comments.